# A Multisensor System Embedded in a Computer Mouse for Occupational Stress Detection

**DOI:** 10.3390/bios13010010

**Published:** 2022-12-22

**Authors:** Thelma Androutsou, Spyridon Angelopoulos, Evangelos Hristoforou, George K. Matsopoulos, Dimitrios D. Koutsouris

**Affiliations:** 1Biomedical Engineering Laboratory, National Technical University of Athens, 15772 Athens, Greece; 2Laboratory of Electronic Sensors, National Technical University of Athens, 15772 Athens, Greece

**Keywords:** occupational stress, stress detection, physiological parameters, cognitive performance, photoplethysmography, galvanic skin response, multisensor

## Abstract

Occupational stress is a major challenge in modern societies, related with many health and economic implications. Its automatic detection in an office environment can be a key factor toward effective management, especially in the post-COVID era of changing working norms. The aim of this study is the design, development and validation of a multisensor system embedded in a computer mouse for the detection of office work stress. An experiment is described where photoplethysmography (PPG) and galvanic skin response (GSR) signals of 32 subjects were obtained during the execution of stress-inducing tasks that sought to simulate the stressors present in a computer-based office environment. Kalman and moving average filters were used to process the signals and appropriately formulated algorithms were applied to extract the features of pulse rate and skin conductance. The results found that the stressful periods of the experiment significantly increased the participants’ reported stress levels while negatively affecting their cognitive performance. Statistical analysis showed that, in most cases, there was a highly significant statistical difference in the physiological parameters measured during the different periods of the experiment, without and with the presence of stressors. These results indicate that the proposed device can be part of an unobtrusive system for monitoring and detecting the stress levels of office workers.

## 1. Introduction

At a time when the rapid development of technology and new forms of global economy have brought about major changes in the nature of the workplace and the conditions prevailing in it, the effective management of occupational stress remains an open challenge. According to the European Agency for Safety and Health at Work, stress is one of the most common work-related health problems [1]. Concurrently, many of the other most-cited health issues caused or intensified by work, such as fatigue, headaches and muscle problems, can often be related to exposure to a stressful environment. More than half of all workers report that stress is common in their workplace, while 60% of all lost days can be attributed to stress and psychosocial risks [2]. Long-term exposure to occupational stress can have serious effects at a social and mental level, such as musculoskeletal disorders and work-related depression, but also at an economic level, as the cost of the latter alone in Europe is estimated at 617 billion euros per year [3].

The COVID-19 pandemic has induced significant changes in the social, economic and labor sectors. Safety and health in the workplace have been affected on many levels. More than four out of ten respondents across the EU agree that their work stress has increased as a result of the pandemic [1]. In order to minimize the spread of the disease, remote working has been significantly enhanced, while the digitization process has been accelerated. The nature and organization of work, for a large proportion of workers, has changed radically with the use of new digital technologies outside the traditional workplace. Many of these changes were maintained even after the lifting of the strict health and social measures enforced due to COVID-19, creating a new working reality. While remote working can offer flexibility and greater autonomy and is considered to promote a better work-life balance, factors such as workplace isolation, family disruption, the absence of peers and a lack of organization can be inhibiting factors in this direction [4]. The aforementioned facts are particularly relevant for office workers, as office automation technologies enable the execution of their work remotely, using computer and communications support. Thus, in the present circumstances, where the traditional office working environment is changing, it is crucial to adapt existing methods of monitoring work-related stress and to adopt systems that can effectively detect and report mental health conditions and issues of employees.

The field of Affective Computing, introduced by Rosalind Picard in 1995 [5], includes a large variety of studies aimed at automatically detecting the emotional state and stress levels of individuals. To achieve this, these studies process and analyze physiological measurements such as heart rate, skin resistance and electrical brain activity as well as behavioral measurements such as facial expressions, body posture and speech variation. The former are often considered more reliable and indicative, while the latter can be measured in a less invasive way.

Regarding the field of the workplace, the need to develop systems for the detection and continuous monitoring of stress, which are as minimally obtrusive as possible, has emerged in the literature in recent years [6,7]. This is based on the fact that sensors and devices worn on the body or attached to it can disrupt users’ routines, work patterns and natural behavior. In addition, social and privacy aspects must always be considered. As a result, there is growing research interest in the utilization of devices used daily in the workplace in order to monitor the well-being and performance of workers, through the use of sensors and novel data analysis methods. The personal computer and its peripheral units are among the prime candidates for the implementation of such systems, as they have become an integral part of office work, helping to reduce workload and increase efficiency [8].

In this study, we present a system for detecting occupational stress in office workers through the analysis of physiological signals from a multisensor system embedded in a computer mouse. Research work, which has been carried out in the past, involves the placement of sensors on the computer mouse for different purposes. Lin et al. [9] attached multi-photoplethysmography (PPG) sensors on the mouse’s surface and applied a weighted average method to accurately detect signal peaks, suggesting this tool could be used for the early detection of pathologies, such as cardiovascular diseases. Tran and Chung [10] designed a wireless PPG mouse with a Bluetooth mote and proposed a robust peak detection algorithm to address motion artifacts. In [11], heart rate variability features were extracted from PPG surfaces mounted on a computer mouse. The aim of this study was to estimate mental stress during office work, but, as far as we are aware, the device was not validated in this context. In [12], a mockup design of an in-house computer mouse with embedded sensors measuring heart rate, skin conductance, skin temperature and grip force was presented. The computer mouse was proposed as part of a project that aimed to help older adults stay motivated in the workplace. A probabilistic algorithm for classification and a use case scenario were described but no future work was found on our part to validate them. On the other hand, Kaklauskas et al. [13] developed and tested a biometric computer mouse advisory system to compute the work productivity and level of emotional state of users. The computer mouse that was designed included sensors measuring skin conductance, heart rate, touch intensity, temperature and humidity of the user’s hand. An overview of the above studies is presented in Table 1.

The proposed system of this study, first introduced in [14], aims to be a simple and easy-to-use device that monitors and detects office occupational stress and can easily adapt to the new working norms, induced by the transformation of the traditional work environment. An experimental protocol involving the execution of tasks simulating a stressful office work environment was designed, developed and implemented to validate the system. To the best of our knowledge, no system with a similar structure that combines noninvasive and user-friendly design with the advantages of wireless data transfer has been documented in the literature so far. Furthermore, there have been no adequately described validation procedures for similar devices implemented for stress detection, through the execution of experimental protocols specifically designed to simulate real-life occupational stressors. Section 2 describes the system architecture, signal processing methods, experimental protocol and procedure along with the statistical analysis tools that were used. The results of the physiological signals analysis and the protocol validation are presented in Section 3. Finally, a discussion of the results and conclusions is provided in Section 4.

## 2. Materials and Methods

### 2.1. System Architecture

The proposed structure of the smart computer mouse introduced in this study incorporates a PPG sensor, a galvanic skin response (GSR) sensor and a development board including a microcontroller and a Wi-Fi module. All the components are integrated in a commercially available, wired optical mouse. The purpose of the system is to determine the stress level of users through the analysis of the physiological parameters derived from the signal processing.

During the use of the system, the signals are recorded by the sensors and sent to the microcontroller. There they are processed and filtered with the use of algorithms for noise and motion artifacts mitigation. Finally, the results are sent to the cloud backend of the development board, through the Wi-Fi module. A mutual authentication using RSA public–private key pairs is used for the communication between the device and the cloud. Moreover, an encrypted session using AES over TCP is created for data transmission. Using the available communication features, data are sent to our external server over TLS/SSL for further analysis.

#### 2.1.1. Hardware Components

The PPG sensor that was selected for this study is the Pulse Sensor by World Famous Electronics LLC (New York, NY, USA). This module includes a green LED, a Broadcom APDS-9008 ambient light sensor and the appropriate electronic components for signal filtering and amplification, soldered on a circular Printed Circuit Board (PCB). Its operation is based on the transmission of light through the user’s skin, which is then received from the light sensor. The changes in the received signal correspond to the blood pulse. The sensor’s output signal is then passed through a passive RC filter network for noise reduction, followed by an operational amplifier, where the signal is amplified and level-shifted in order to use half of the supply voltage as a voltage reference. As a result, the final output can be fed to a microcontroller for further processing.

The GSR sensor that is used is the Grove-GSR sensor by Seeed (Shenzhen, China). This module includes a PCB board where two disc-shaped electrodes are connected. This sensor is able to measure variations in skin resistance through the two electrodes that are in contact with the user’s skin. Then, these resistance variations can be associated with emotional variations.

As shown in Figure 1, both the PPG and the GSR sensors are connected to a Particle Photon (San Francisco, CA, USA) development board. This board includes a 120 MHz ARM Cortex M3 microcontroller, as well as a Broadcom Wi-Fi chip for wireless communication. The board’s recommended supply voltage is in the range of 3.6–5.5 V, enabling its supply through a standard USB port. By calculating the total energy consumption, it can be shown that it is possible to power both the mouse and the integrated parts (sensors and development board) through the existing mouse cable. More specifically, the current consumption of the mouse is equal to 100 mA, Particle Photon’s (Wi-Fi enabled) current consumption equals 80 mA, PPG sensor’s is 4 mA and GSR sensor’s is 1.5 mA. Hence, a typical USB port of 5 V and 500 mA is sufficient for the project. The placement of the different components inside the device and the connections among them are shown in Figure 2a.

The mouse that can be a suitable option for hosting the sensors was selected based on some requirements. Its interior needed to be large enough to enclose the sensors, their electronics and the development board. Moreover, its external surfaces had to be flat enough to incorporate the sensors without affecting its usability. Finally, an intended feature was its wired communication with the computer, in order to provide sufficient and uninterruptible power to the components.

The main goal of the sensor’s placement was to develop a device able to successfully conduct the desired measurements, without affecting the user’s actions. Both the development board and the GSR sensor’s electronics were placed inside the mouse, while the PPG sensor was mounted on the side wall of the mouse at the thumb’s location, where an appropriate hole was drilled. The two GSR sensor’s electrodes were placed on the top surface of the mouse where the user’s palm rests. The exterior of the custom device is shown in Figure 2b.

When the mouse is connected to a computer’s USB port, the Particle Photon is automatically powered on and scans for any available Wi-Fi network. If an already known Wi-Fi network is found, it automatically connects to it, using the stored Wi-Fi credentials. After the establishment of the connection, the sensors’ data are acquired and sent over the network to the desired database.

#### 2.1.2. Software Components

Particle Photon can be programmed remotely through the Particle Integrated Development Environment (IDE), using a typical web browser. This feature is especially useful in this particular project as it enables us to update the mouse’s code (if necessary), without disassembling it. Moreover, it provides a web-based dashboard where critical information about the status of the board (e.g., Wi-Fi SSID, signal strength and latency, incoming data values, logs and warnings, etc.) is shown in real time.

The developed algorithm includes all the appropriate libraries and functions that are required for the proper functionality of the sensors, in order to read sensor outputs, preprocess the raw data of the signals, and finally, transmit them through the cloud. Moreover, it provides the ability to adjust certain values, such as measurement intervals, filter parameters, etc.

As a result, measurements from both the PPG and the GSR sensors are received by the microcontroller and sent wirelessly through an encrypted connection. Afterwards, the received data can be postprocessed to decrease measurements’ noise and artifacts. Finally, the resulting data are stored in a local database for further analysis.

### 2.2. Signal Processing

The GSR and PPG signals were acquired by the system sensors with a sampling frequency of 500 Hz and transferred to the microcontroller. During the data acquisition, filtering and preprocessing techniques were applied to mitigate the noise factor and extract the necessary values for subsequent processing.

From the PPG signal, we attempted to detect successive events of instantaneous heartbeats and measure interbeat intervals. In this process, a key aspect for accurate results is to identify the exact moment when each heartbeat occurs. Baseline noise and false readings due to the dicrotic notch can often interfere with reliable beats-per-minute (BPM) calculation. In our case, this is addressed by applying an algorithm that calculates interbeat intervals by measuring the time between the moments when the signal crosses 50% of the wave amplitude during its rapid rise [15]. Simultaneously, time thresholds are used to avoid high-frequency noise and interference from the dicrotic notch.

Another challenge is the fact that the PPG signal is quite sensitive to motion and ambient light variations during recording [16]. In general, the successful removal of noise and motion artifacts in PPG signals has been the subject of many research studies in recent years [17]. One of the proposed methods, which, however, has not been extensively studied as a PPG noise removal method yet, is the use of the Kalman filter [18]. Kalman filtering aims to reduce the noise of the measured sensor value by adjusting it based on the past sensor data. For this purpose, it follows a procedure that can be divided into two parts: prediction and update [19]. In the prediction part, the algorithm calculates the next sensor measurement based on the past data. During the update process, the prediction value is refined based on the measured value in order to approximate the actual one. In this study, we applied a simple one-dimensional Kalman filter to eliminate motion artifacts due to the computer mouse movement.

The GSR signal is also affected by artifacts such as body gestures and movements or improper contact between the electrodes and skin [20]. In the proposed experimental setup, the electrodes are not attached to a wearable device but are instead embedded in a surface that is utilized daily at work, i.e., a computer mouse, which regularly moves with the user’s hand. Thus, we applied a moving average filter to smooth the data and reduce the signal artifacts. Subsequently, we converted the voltage output (Vout) of the sensor to human skin conductance (SC) measured in Siemens, by adapting the conversion formula provided by the sensor manufacturer [21] to our system’s specification, resulting in Equation (1):(1)$$SC=\frac{2048-V_{out}}{\left(4096+2\cdot V_{out}\right)\cdot 10000}$$

### 2.3. Experimental Process

A series of stress-inducing tasks requiring the use of a computer keyboard and the proposed custom computer mouse was designed and developed in this study. Specifically, the execution of each task either involved the use of the computer mouse from which physiological parameters related to PPG and GSR signals were obtained, or involved the use of the keyboard from which behavioral parameters related to keyboard dynamics were obtained, or required their combined use in the same time interval. For the purpose of this paper, our analysis focuses on the processing of the physiological parameters recorded through the use of the computer mouse. The design of the protocol allows us to analyze individually the physiological parameters and their contribution to stress in an office environment, while concluding the validation of the custom device that is proposed. The following sections describe the aforementioned methodology in detail.

#### 2.3.1. Participants

Thirty-two participants between 20 and 40 years of age (M = 29.34, SD = 4.65, 12 women and 20 men) were included in the study. Four of them stated that they have a moderate level of familiarity with computers, while thirteen and fifteen stated that they have a high and a very high level of familiarity, respectively. All the subjects use the computer daily as part of their work or their studies. Three of the subjects declared that they had participated in a similar experiment in the past. The participants were not selected based on any assessment criteria regarding their stress levels. The only condition for their participation was that they had to use the computer mouse with their right hand, due to the design of the experimental setup. Prior to the experiments, written and verbal information explaining the procedure in detail were given, and written informed consents were obtained. However, the stress-induction aim of the study was disclosed after the completion of the experiment to avoid bias in the measurements and results.

#### 2.3.2. Experimental Protocol

In the present study, we sought to simulate an office work environment in a regular state and in the presence of stress. For this purpose, existing laboratory stressors were used, slightly modified to serve both the setup of the experiment and to be more compatible with real-life scenarios. As working in an office environment always involves a certain amount of mental strain, we chose to distinguish between a condition of concentration and mild mental load and a condition loaded with work stressors, as proposed by [22]. These states are referred to hereafter as the control condition and the stress condition. Therefore, each task included in the experiment was divided into two levels based on the above-mentioned conditions. The design of the protocol considers the predominant stressors of work, namely mental load, time pressure and social pressure related to both performance and the threat of evaluation by other people [23]. To challenge the threat of appraisal, participants performed some of the tasks in front of an audience. Specifically, during the stress-inducing level of each task, the experimenter took a position next to the participant, visually accessing the test interface on the computer screen and taking notes. In contrast, during the control level, the researcher sat at a distance from the participant and did not participate in the experimental process. The experimental protocol of the study included four tasks, two of which are within the scope of this paper, as they exclusively involved the use of the computer mouse. Their design is described in detail in the following sections.

##### Stroop Color Word Task

Stroop Color Word Test is a neuropsychological test named after J. R. Stroop that is used to assess the ability to inhibit cognitive interference during the simultaneous processing of two different stimulus attributes [24]. The test procedure involves displaying colored words to the subjects and asking them to name the color of the word rather than read the word itself. Its effect is based on the delay in reaction times between congruent and incongruent stimuli. The congruency occurs when the meaning of a word and its font color are the same. We used two versions of the test with distinct levels of difficulty in order to distinguish between the control condition and the stress condition. In the first version, a word appears in the center of the computer screen. Below the colored word, there are three buttons with candidate answers. The participants are asked to select the correct answer by clicking on the corresponding button. The latter turns green or red, informing them whether they have given the correct or wrong answer, respectively, before moving on to the next word. At the top right of the screen, the score is displayed and updated after each answer. The first level of the test mainly involves congruent stimuli, where the colors and the word names are matched. In contrast, in the second level, the words and colors are incongruous, and the subjects must click on the correct option within three seconds. A countdown timer is placed at the top left corner of the screen. In this way, an attempt is made to intensify the cognitive interference, while introducing time pressure as a stressor. An example of the task’s interface can be seen in Figure 3a.

##### Mental Arithmetic Task

Mental Arithmetic tests have been widely used as stressors, and are often included in well-known experimental protocols such as the Trier Social Stress Test [25]. In the first level of the experimental procedure of the present study, participants were asked to perform mathematical calculations involving the four basic mathematical operations, namely addition, subtraction, multiplication and division. Each time, a mathematical operation between two numbers up to two digits is displayed on the computer screen, where either one of them or the result of the operation needs to be completed. The participants enter their answer and submit it by clicking on the appropriate button. The color of the numbers turns green or red, informing them whether they answered correctly or incorrectly, before moving on to the next mathematical operation. The performance score, which increases with each correct answer and decreases with each incorrect answer, is displayed at the top right of the computer screen. In the second level, which concerns the stress condition, the same procedure is repeated. However, at this level there is a time limit of ten seconds for each answer to be submitted. If the time, displayed via a countdown timer, is exceeded, an automatic transition to the next mathematical operation is made and the score decreases. The mathematical operations of both levels are generated by means of random integer generators. However, the ranges used to generate the numbers in the second level are set in such a way that the complexity and difficulty of the calculations is increased significantly, intensifying the mental load. An example of the task’s interface can be seen in Figure 3b.

#### 2.3.3. Experimental Procedure

The experimental setup attempted to simulate a typical computer workspace (Figure 4). Upon arrival, the researcher enacting as the experimenter provided information to the participants to familiarize them with the experimental equipment and sensors. When the briefing and collection of the consent form were completed, the main experimental procedure began. First, participants answered some general questions. Then, they performed the two levels of each task of the experimental protocol. After each level, they answered a self-report questionnaire. Resting periods were introduced at the beginning and end of the procedure, as well as between the different tasks. The role of these periods was firstly to ensure that the measurements were not affected across the different condition setups and secondly, to obtain the baseline signal of each subject.

The entire experimental procedure was carried out through a web application implemented on the Flutter framework, that ran locally on the computer of the experimental setup. The mapping of the recorded signals on the different stages was enabled by automated event marking. The schematic diagram of the procedure is shown in Figure 5. The different experimental conditions are described in detail below:

##### General Questionnaire

The questionnaire included questions about demographic characteristics such as gender, age, employment and questions about computer familiarity and use. Subjects also answered questions related to factors that could influence the measurements of an experiment designed to study stress, such as caffeine and alcohol consumption, medication intake and participation in a similar experiment in the past.

##### Resting Periods

During the resting periods, videos with images of natural landscapes and animals were shown. The videos were accompanied by relaxing music. The first video of the experimental procedure, before any task was performed, was three minutes long, while the remaining videos between the different tasks and at the end of the experiment were two minutes long. Throughout these periods, subjects were required to watch the videos in a comfortable seated position, limiting their movements as much as possible and with their right hand resting on the mouse, so that their physiological parameters be monitored.

##### Self-Report Questionnaire

NASA Task Load Index (NASA-TLX) was used as the self-report questionnaire [26]. The questionnaire includes six questions, concerning the mental, physical and temporal demand, performance, effort and stress/frustration. The answer options are on a scale of 1 to 10. The questionnaire was filled in after both levels of each task, resulting in a total of eight answered questionnaires for each participant.

##### Stress Tasks

The tasks described in detail in Section 2.3.2 were used to induce stress in the participants. Each task consisted of two levels designed to distinguish between the control condition and the stress condition setup after the addition of the selected stressors. Before the beginning of each task, subjects received detailed instructions on the computer screen, while at the end of each task, they were informed about their performance before moving on to the next step. The order in which each participant had to perform the different tasks was not fixed. In each individual task, however, the levels were always presented in the same order, with the stress condition following the control one.

**Figure 5 biosensors-13-00010-f005:**
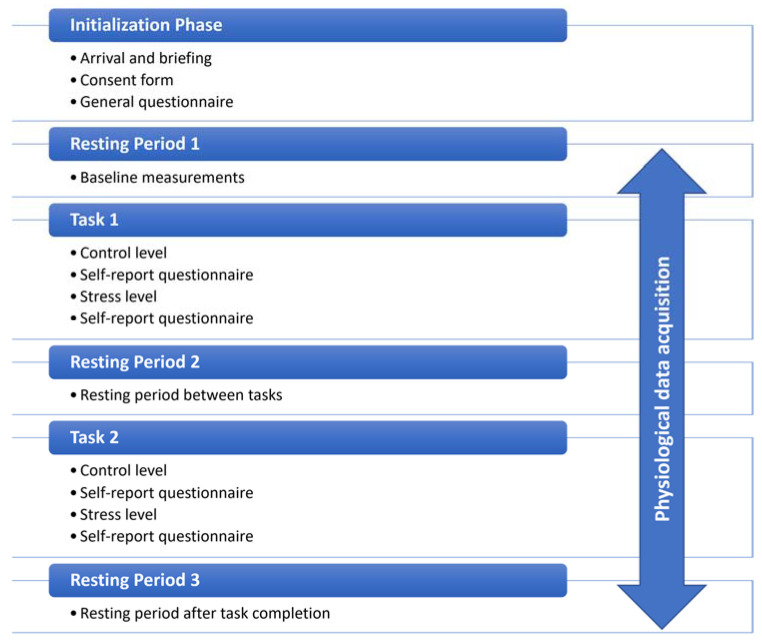
The schematic diagram of the experimental procedure.

### 2.4. Statistical Analysis

Statistical analysis methods were applied to validate the experimental protocol that was used and verify the distinction between the control and the stress condition through the results obtained from the signal processing. The significance level we chose for all the cases of the statistical tests was 0.001. First, the Lilliefors test for normality [27] was applied to the parameters for analysis. The test showed that the values did not follow a normal distribution, thus nonparametric statistical tests were used for further analysis.

The protocol used in this study was designed to induce stress in participants using stressors that occur in the office workplace. In order to validate the experimental protocol, we analyzed the questionnaire answers regarding the last question, where subjects indicated their stress level in the different phases of the process. The Wilcoxon signed-rank test [28] was applied to indicate if there was a significant statistical difference between the control and stress states, both for all the stress tasks as a whole and individually for each of them.

We then examined how much the different phases of each task of the experiment affected performance. For this purpose, we compared the subjects’ scores that were computed according to the scoring system described in the previous sections. As each task had a different maximum score that the participants could achieve, we standardized the calculated scores to fit them to the same scale, with a maximum value of 100, in order to make it easier to compare performance across tasks. Moreover, we examined the users’ answers to the question of the self-report questionnaire regarding their performance. The Spearman’s rank correlation coefficient [29] was calculated between the answers regarding the score and the performance to examine whether subjects’ perception of their performance affected their stress levels.

Regarding the analysis of the signal processing results, the null hypothesis for our application was that there is no statistically significant difference between the measurements of physiological parameters performed in the control and stress periods for both tasks. The datasets generated for the two levels of each task were unbalanced, as there was no fixed time limit for completing each level during the design of the experiment. In addition, the values were not normally distributed. As a result, we applied the Mann–Whitney U statistical test [30] to the data and investigated variations in the features of BPM and SC that were calculated from the PPG and GSR signal during the differentiated conditions.

## 3. Results

PPG and GSR signals were recorded for all participants during the experiment, leading to the creation of 32 datasets. BPM and SC values were calculated for the two levels of each task, as well as for the resting periods recorded before, after and between the tasks. Figure 6 shows the range in which the mean values of the participants’ physiological parameters varied during the Stroop Color Word Task and the Mental Arithmetic Task. It can be observed that the range of the mean values of the parameters is large, as is the variation, especially during the stress periods of the two tasks. The median of the mean values in the case of BPM is higher during the stress conditions, while in the case of SC, changes are observed in terms of skewness of the data.

### 3.1. Signal Processing

This section presents an example of applying the signal processing methods to the physiological parameters of a participant during a specific time period of the experiment. It is worth noting that the choice of the time window was arbitrary and is not intended to show the evolution of values in any of the resting, control or stress states.

The results of the application of Kalman filtering on BPM data derived from the PPG signal are shown in Figure 7. During the filtering process, special attention was paid to the determination of appropriate input variables. Since the sensor equipment used in the experimental setup did not provide any specific noise information, the parameters related to the noise and model measurement variance were determined with real data. More specifically, by providing an input signal of known frequency to the PPG sensor, we were able to quantify its noise levels and determine the appropriate input parameters to ensure optimum filter performance.

The results of the GSR signal processing procedure on experiment measurements are shown in Figure 8. The moving average filter was applied on a window size of ten measurements.

### 3.2. Protocol Validation

The boxplots of self-reported stress levels for all subjects are shown in Figure 9a. By observing this figure, it is evident that the reported levels during the stressful conditions of the experiment are on average considerably higher than the stress levels during the control periods. In order to validate this, a statistical analysis was performed on the reported stress levels. First, the Lilliefors test showed that the values did not follow a normal distribution, as expected, as they were derived from responses to a specific question of the questionnaire and the answer options were on a scale of 1–10. As it can be seen from the results of the Wilcoxon test that are shown in detail in Table 2, the *p*-value for all cases of the test was less than 0.001, which was the significance level we chose. The fact that the W value of the test equals zero in all cases means that none of the participants rated the stress experienced in the first phase of the tasks higher than in the second.

The boxplots in Figure 9b show the participants’ answers to the question of the self-report questionnaire regarding their performance. We notice that for both tasks the reported level of performance is lower in the stress condition than in the control condition. However, while in the Stroop Color Word Task the level values are quite close, in the Mental Arithmetic Task there is a large discrepancy. This is supported by Figure 10, which shows the average scores of all participants as calculated during the experiment. Thus, it is evident that users’ perception of their performance is in line with the scores calculated during the task.

We expected that the perceived stress levels and performance, according to the subjects’ answers, would be negatively correlated. This was confirmed by the Spearman correlation coefficient that indicated a significant negative correlation of −0.58 (*p*-value = 6.93 × 10^−13^).

### 3.3. Physiological Parameters Analysis

Figure 11a,b display the changes in the values of BPM and SC during the experimental procedure for two subjects A and B, respectively. All resting, control and stress periods have been delineated and labeled. The periods when the subjects completed the self-report questionnaires after each level of each task are also marked.

The evolution of SC values over time leads to the observation that in most cases the reaction to the onset of an experimental task causes them to increase rapidly. This change may remain throughout the duration of the task (section S2 in Figure 11b) or start to decay after a short interval (section C1 in Figure 11a). During resting periods, the conductivity generally shows slighter changes. By observing the evolution of BPM, we find that in the case of subject A, there is an increase in values, which escalates during the transition from control levels to stress levels and fades during resting periods. On the other hand, in the case of subject B, the values show greater variations, while the increase in pulse rates during stress periods is often maintained during the periods of questionnaire completion and the resting periods that follow them.

The results of the statistical tests applied for BPM and SC are presented in Table 3. The analysis was performed for all the stages of the three different phases of the experiment, i.e., resting, control and stress periods. We observe that for the BPM values, the null hypothesis can be rejected for the Mental Arithmetic Task but not for the Stroop Color Word Task. Moreover, no highly significant statistical difference emerges, not only between the control periods (C1, C2) and between the stressful phases (S1, S2) of the two tasks but also between phases of different categories (C2, S1). In addition, significant differences do not arise for BPM values between some resting periods and periods when the tasks are performed. The latter is also observed for GSR values, which, however, show a highly significant statistical difference between the control and stress periods of each task, rejecting the null hypothesis.

## 4. Discussion

The aim of this study was to validate the use of a custom device, consisting of a multi-sensor system integrated in a computer mouse, in the detection of occupational stress in an office environment. To achieve this, an experimental protocol was designed and implemented based on well-known, and widely used in the literature, stress-inducing tasks, while adapting it, where necessary, to accommodate the experimental setup and implementation context of the proposed system.

The experimental procedure was designed to simulate a typical workplace by applying the stressors most frequently encountered there. The analysis of the users’ responses to the reference questionnaires shows that the intervention achieved its objective. There was a highly significant statistical difference in reported stress levels between the control condition period and the stress condition period, both for each task individually and for the two tasks as a whole. The Mental Arithmetic Task was reported to be considerably more stressful than the Stroop Color Word Task. This may have been attributed to the difference in the difficulty level of the stress periods of the two tasks. The fact that the arithmetic operations in the Mental Arithmetic Task were generated with random integer operators sometimes led to the appearance of intractable problems at the second level, considerably more difficult than those at the first level. This further resulted in a variety of challenges that subjects had to face during the task, unlike the Stroop Color Word Task, where the habituation effect was more likely to occur.

The above were also reflected in cognitive performance, as subjects’ scores were much lower in the stress period of the Mental Arithmetic Task. The correlation analysis showed that there is a significant negative relationship between the reported stress levels and the perceived performance level of the individuals, indicating that when employees perceive that they are not performing well in their assigned tasks, they experience increased levels of stress.

GSR and PPG signals were collected from 32 subjects participating in the experimental procedure. From their processing and analysis, the physiological BPM and SC values were extracted and the possibility of detecting work stress was studied. A review of the evolution of these parameters over time showed that under stress, both BPM and SC increase. However, it is evident that there are inter and intra-subject differences in response to stress-inducing tasks. The impact of the response on physiological measures depends on the magnitude of the arousal and each subject’s perception of the specific threat. There are instances where subjects are on continuous high alertness during the presence of a stressor stimulus and others where they exhibit brief responses.

Preprocessing and filtering techniques were applied to eliminate noise artifacts. The moving average method is often used to preprocess GSR signals [20,31]. In this study, we applied the 10 point moving average filter that smoothed the signal, adequately eliminating noise from the motion and the environment. Moreover, the application of Kalman filtering sufficiently mitigated the PPG signal artifacts, caused by motion and ambient light noise. This suggests that it may be a promising tool in PPG signal processing and its applicability needs to be further studied.

Statistical analysis was carried out on the calculated physiological parameters. The null hypothesis was that there is no statistically significant difference between the measurements obtained during the control and stress periods for both tasks. The analysis was performed using statistical tools for each parameter individually.

SC values showed highly significant statistical differences between the control and stress phases for both tasks of the experiment. This confirms that it is one of the most reliable physiological measures for stress detection [32]. BPM values showed highly significant statistical differences between the two phases of the Mental Arithmetic Task but not between the corresponding phases of the Stroop Color Word Task. This may be due to the fact that the latter involved much longer and abrupt movements of the computer mouse during its execution. Therefore, the use of the Kalman filter applied to eliminate motion artifacts may have affected the useful information that could be extracted from the rapid or more subtle changes in the subjects’ signals. In future work, we will test alternative values for the Kalman filter parameters that better serve the purposes of the intervention. Moreover, the use of the system in other stressful tasks could lead to additional useful conclusions.

The values of the physiological parameters during the resting periods exhibited in many cases a significant (*p*-value < 0.05), but not highly significant, statistical difference with the values of the task periods of the experiment. This mainly concerns the resting periods following the application of the stressors and can be explained by the fact that the subjects’ physiological signals did not have sufficient time to fully recover after the induced stress and approach the baseline values.

There are several inhibiting factors that can affect the measurements and hinder the efficiency of stress detection. For example, 56% of the participants responded that they had consumed caffeine within a five-hour period prior to the experiment, while 15% of them stated that they were taking some kind of medication. These, together with other personal characteristics and conditions, such as the presence of arrhythmias, should be taken into account when interpreting the results. In addition, while care was taken to ensure that the experiments were performed in a specific time frame and under the same conditions during the day, differences due to variations in environmental conditions cannot be excluded. Furthermore, in order to investigate the gender aspect, we applied the analysis described in Section 2.4 separately to male and female participants. The results that emerged led to the same conclusions. However, minor changes were observed suggesting that different tasks may affect each gender to a different degree. In our case, the Stroop Color Word Task seemed to be more stressful for male participants. Nonetheless, more measurements and systematic analysis is needed to draw conclusions about the influence of stressors in relation to gender.

Occupational stress is a known health risk for multiple diseases and disorders, thus its automated detection through the analysis of physiological and behavioral parameters has been the subject of a large number of studies. In recent years, the need to develop unobtrusive systems that can be easily adapted to new forms of workplaces has become increasingly evident. The presented system aims to be a plug and play device that can be easily used by office and remote workers. The multisensor system that is fully integrated into a computer mouse can reliably record the user’s physiological signals and transmit them wirelessly. Analysis of these signals and extraction of appropriate parameters proved to be effective in monitoring the users’ condition and detecting their stress levels while working on the computer. To our knowledge, a similar system has not been proposed in the literature that has a noninvasive structure, transmits data wirelessly and has been tested in an appropriately designed experimental procedure that includes the main job stressors. Future steps are to extract features from physiological signals and combine them with features of behavioral measures derived from computer mouse and keyboard use in order to create a reliable multimodal detection system. Moreover, a comparison with similar systems based on unobtrusive smart wearable devices [33] will be performed in order to draw conclusions about the specifications of an efficient system for monitoring and detecting stress levels in the workplace.

## 5. Conclusions

This study developed and proposed a custom device consisting of a PPG and GSR sensor system embedded in a computer mouse. The biosignals recorded during the use of the device are transmitted wirelessly through an integrated development board with a Wi-Fi module. The system was validated for its application in automatic stress detection in an office environment by implementing an experimental protocol using stress-inducing tasks with common work stressors. Motion and noise artifacts were mitigated by applying Kalman and moving average filters. The Kalman filter proved to be effective and promising for PPG signal preprocessing; however, further research needs to be conducted to investigate the parameters that achieve the balance between noise elimination and the preservation of useful information for stress detection. The reported stress levels of the participants in the experiment increased during the stressful tasks, while their performance declined, validating the effectiveness of the intervention. Statistical analysis performed on the calculated BPM and SC parameters showed that in the majority of cases there was a highly significant statistical difference (*p*-value < 0.001) between the different periods of the experimental protocol. The proposed system is intended to be part of a noninvasive, easily adaptable and user-friendly system for monitoring and the automatic detection of office workers’ stress levels. In future research, physiological data collected from the device will be combined with behavioral data regarding keyboard and mouse dynamics to develop an improved multimodal system.

## Figures and Tables

**Figure 1 biosensors-13-00010-f001:**
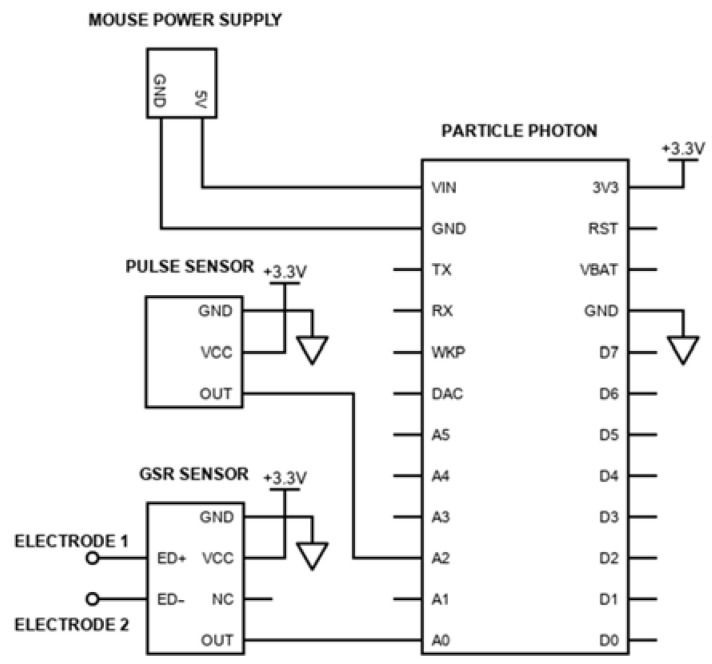
The electrical diagram of the proposed system. The PPG and GSR sensors’ outputs are connected to the analog pins of the development board. The development board is connected to computer mouse’s power pins so that the system is powered by the existing mouse cable.

**Figure 2 biosensors-13-00010-f002:**
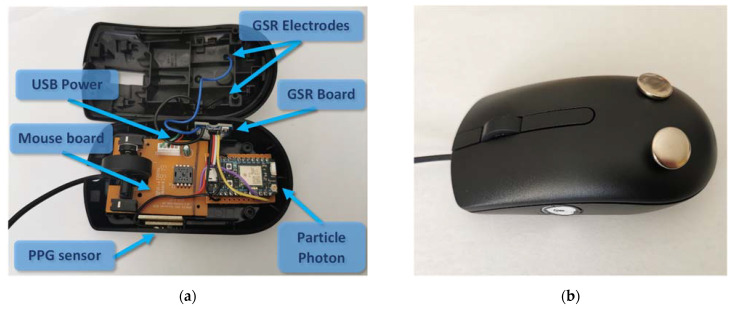
The developed computer mouse with the embedded PPG and GSR sensors, as well as the microcontroller’s board. (**a**) The interior of the device, where the components are indicated; (**b**) the exterior of the final device, where the PPG and GSR sensors are shown.

**Figure 3 biosensors-13-00010-f003:**
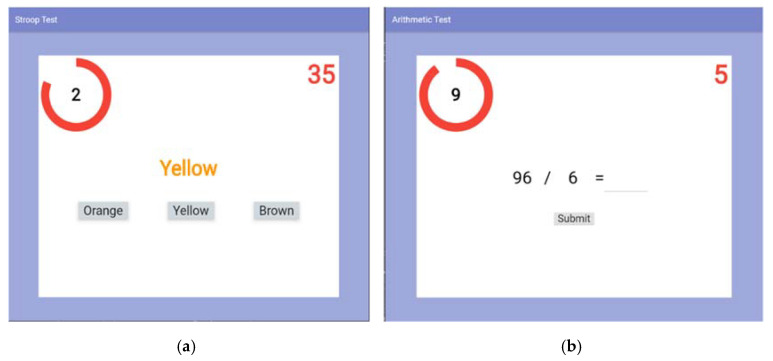
Examples of the interfaces of the experiment’s tasks. The subject’s score is displayed in the upper right corner of the screen. The countdown timer is displayed in the upper left corner of the screen. (**a**) The interface of Stroop Color Word Task. (**b**) The interface of Mental Arithmetic Task.

**Figure 4 biosensors-13-00010-f004:**
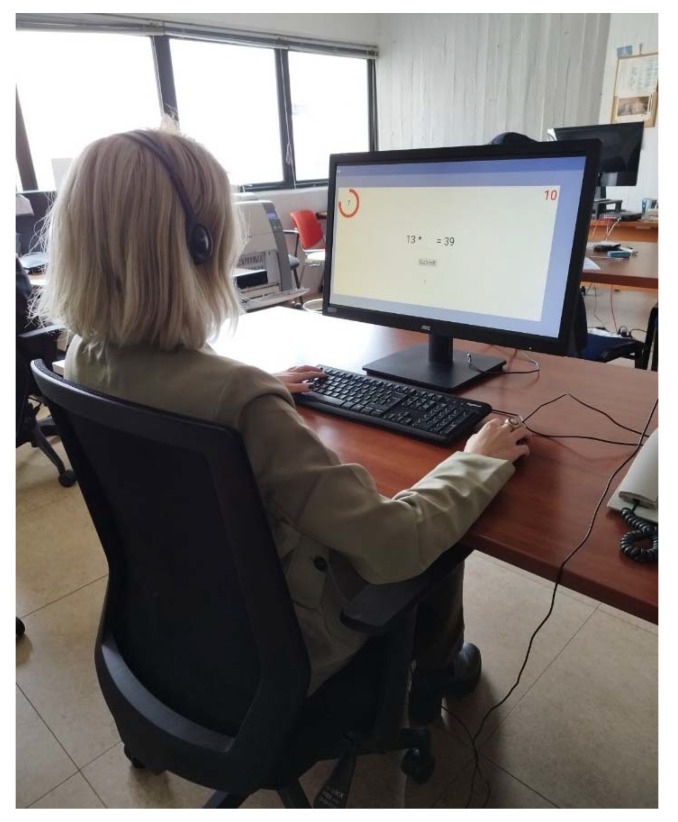
The experimental setup. The subject is seated in an office environment and performs stress-inducing tasks on the computer using the computer’s custom mouse and computer keyboard.

**Figure 6 biosensors-13-00010-f006:**
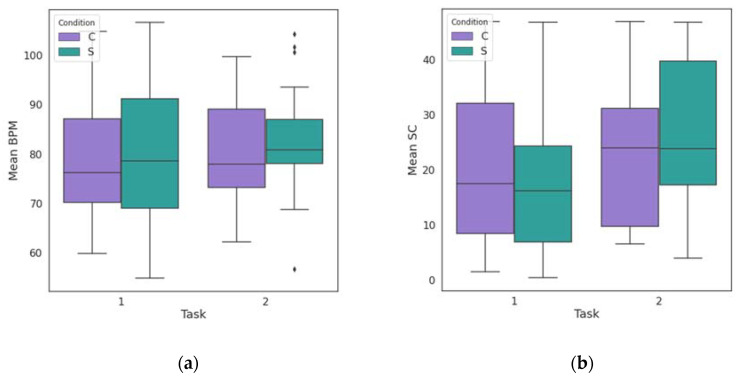
(**a**) Boxplot of mean BPM values of all subjects during the control (C) and the stress (S) conditions of the Stroop Color Word Task (Task 1) and the Mental Arithmetic Task (Task 2). (**b**) Boxplot of mean SC values of all subjects during the control (C) and the stress (S) conditions of the Stroop Color Word Task (Task 1) and the Mental Arithmetic Task (Task 2). The ends of the whisker are set to 1.5 times the interquartile range above the third quartile (Q3) and 1.5 times the interquartile range below the first quartile (Q1). The outliers are shown for each condition by the diamond markers.

**Figure 7 biosensors-13-00010-f007:**
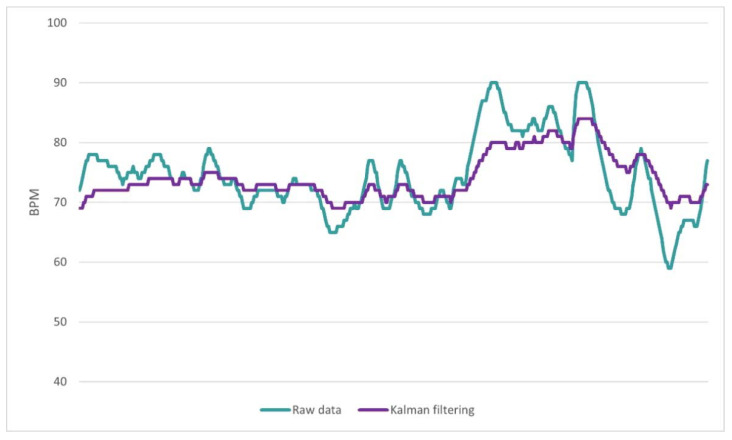
Application of the Kalman filtering algorithm to BPM data that are derived from the PPG signal. The green line depicts the raw data that are obtained from the sensor, before the preprocessing, while the magenta line depicts the same data after the application of Kalman filtering.

**Figure 8 biosensors-13-00010-f008:**
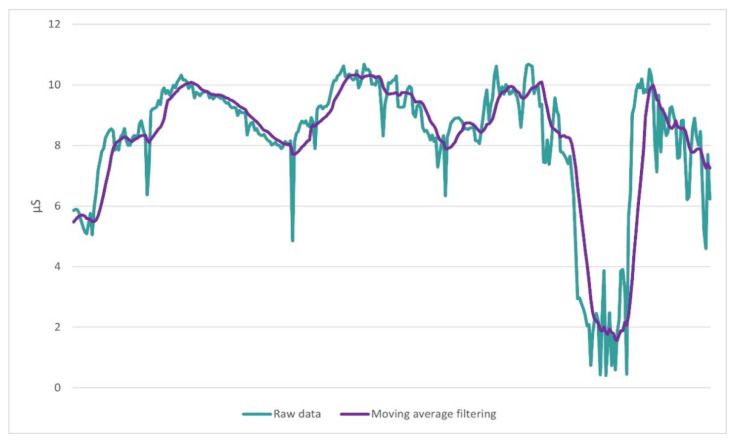
Application of the moving average filtering to the GSR signal. The green line depicts the raw data that are obtained from the sensor, before the preprocessing, while the magenta line depicts the same data after the application of moving average for a period of 10 measurements.

**Figure 9 biosensors-13-00010-f009:**
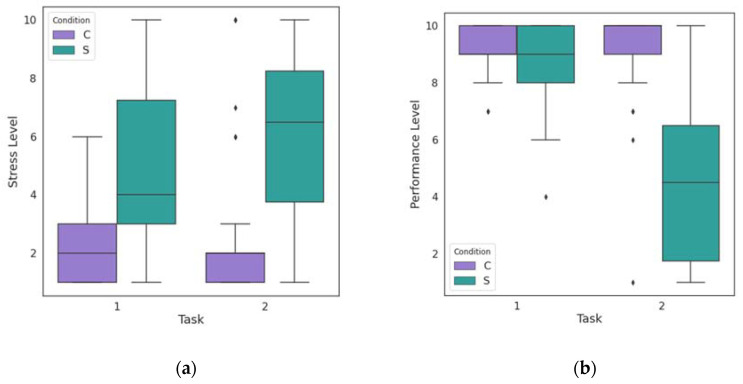
(**a**) Boxplots of self-reported stress levels of all subjects during the control (C) and the stress (S) conditions of the Stroop Color Word Task (Task 1) and the Mental Arithmetic Task (Task 2). The value 1 corresponds to “very low” and 10 to “very high” stress level. (**b**) Boxplots of self-reported performance levels of all subjects during the control (C) and the stress (S) conditions of the Stroop Color Word Task (Task 1) and the Mental Arithmetic Task (Task 2). The value 1 corresponds to “very poor” and 10 to “very good” performance level. The ends of the whisker are set to 1.5 times the interquartile range above the third quartile (Q3) and 1.5 times the interquartile range below the first quartile (Q1). The outliers are shown for each condition by the diamond markers.

**Figure 10 biosensors-13-00010-f010:**
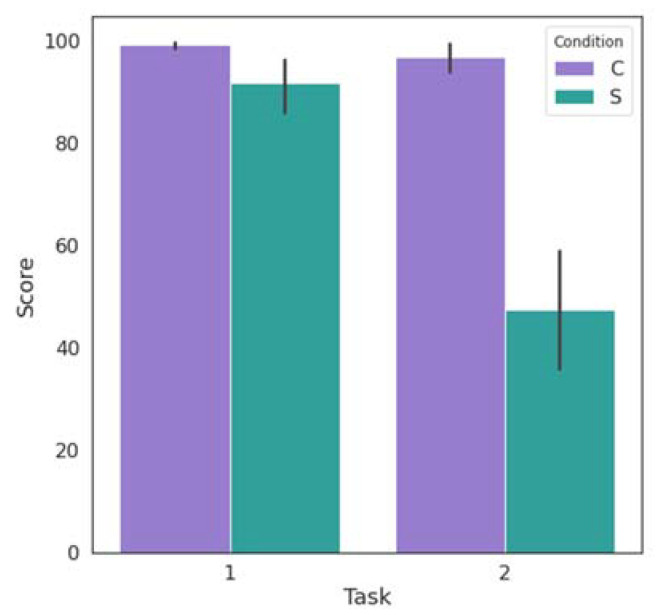
The average scores of all the subjects during the control (C) and the stress (S) conditions of the Stroop Color Word Task (Task 1) and the Mental Arithmetic Task (Task 2). Bar lines indicate the standard deviation values.

**Figure 11 biosensors-13-00010-f011:**
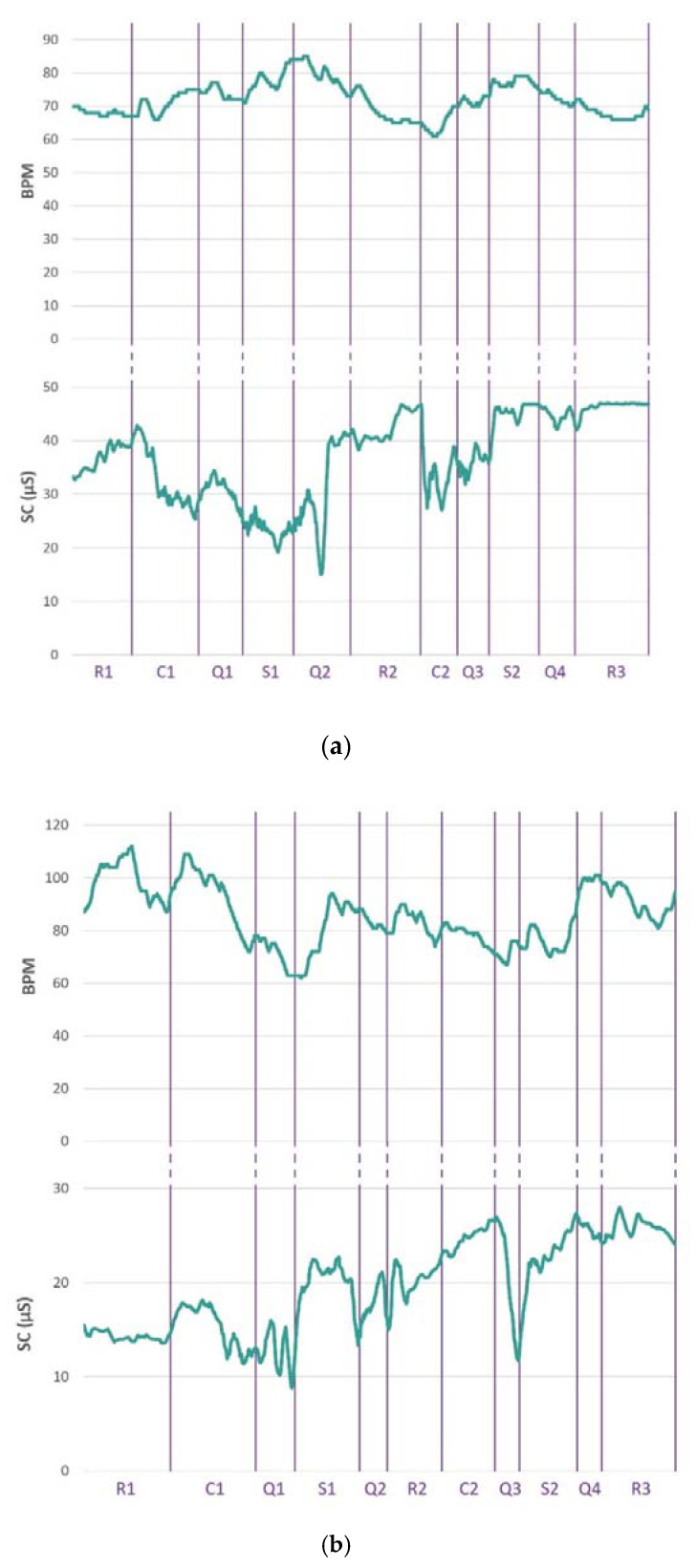
The evolution of BPM and SC values during the experimental procedure. Resting periods (R1, R2, R3), control periods (C1, C2), stress periods (S1, S2) and self-report questionnaire completion periods (Q1, Q2, Q3, Q4) are marked. All periods are presented in the order in which they were performed. The time intervals C1, Q1, S1 and Q2 correspond to the Stroop Color Word Task, while C2, Q3, S2 and Q4 correspond to the Mental Arithmetic Task. (**a**) BPM and SC values for subject A; (**b**) BPM and SC values for subject B.

**Table 1 biosensors-13-00010-t001:** Studies with sensor systems embedded in computer mouses.

Study	Participants	Input	Aim	Methods	Results
[9]	21	PPG	Pulse rate detection	Weighted average method	Sensitivity: 99.71%—rest condition, 94.24%—slow computer mouse movement, 84.36%—rapid mouse movement, 89.04%—browsing
[10]	10	PPG	PPG peak detection	Robust peak detection algorithm	Absolute estimation error: 1.88 beats per minute, maximum average error: 3.1%
[11]	5	PPG	Mental stress monitoring	Signal amplification, filtering and digitization	Correlation coefficients r^2^ between mouse surfaces and point sensor: 0.97 and 0.99, respectively
[12]	–	Heart rate, temperature, grip force, GSR	Stress detection	Bayesian-based probabilistic algorithm	–
[13]	239	Heart rate, temperature, humidity, skin conductance, touch intensity	Detection of stress dependencies on physiological parameters	Linear regression models	Diastolic blood pressure, systolic blood pressure and temperature are predictors of stress levels (*p*-values *→* 0)

**Table 2 biosensors-13-00010-t002:** The results of the Wilcoxon signed-rank test performed on the responses of the self-report questionnaires. W value indicates the sum of the ranks of the differences above or below zero and p-value is calculated under the null hypothesis that the data are not statistically significant. The test was used to compare the control and stress states of each of the experimental tasks. The questionnaire responses for the two states were also summed for both the tasks, so that an overall validation of the experimental protocol could be conducted.

Experimental Task	W Value	*p*-Value
Stroop Color Word Task	0	1.12 × 10^-5^
Mental Arithmetic Task	0	7.80 × 10^−6^
All	0	4.58 × 10^−10^

**Table 3 biosensors-13-00010-t003:** The *p*-values of the Mann–Whitney U tests performed on the BPM and SC parameters that were measured during the resting (R1, R2, R3), the control (C1, C2) and the stress (S1, S2) periods of the experiment. Periods C1 and S1 correspond to the Stroop Color Word Task, while periods C2 and S2 correspond to the Mental Arithmetic Task. The values above the main diagonal correspond to the tests performed on BPM values, while the values below the main diagonal correspond to the tests performed on SC values. The asterisk symbol is placed where *p*-value is smaller than 0.001.

	BPM	R1	R2	R3	C1	C2	S1	S2
SC								
** R1 **			** * **	** * **	** * **	** * **	** * **	** * **
** R2 **		** * **		** 0.27 **	** 0.002 **	** * **	** 0.004 **	** 0.05 **
** R3 **		** * **	** * **		** * **	** * **	** 0.007 **	** 0.45 **
** C1 **		** * **	** * **	** * **		** 0.018 **	** 0.08 **	** * **
** C2 **		** * **	** 0.003 **	** * **	** * **		** 0.006 **	** * **
** S1 **		** 0.04 **	** 0.13 **	** * **	** * **	** * **		** 0.003 **
** S2 **		** * **	** * **	** 0.03 **	** * **	** * **	** * **	

## Data Availability

The data presented in this study are available on reasonable request from the corresponding author.

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
