# Peer review of "A Multisensor System Embedded in a Computer Mouse for Occupational Stress Detection"

_biosensors, 2022, doi:10.3390/bios13010010_

Round 1
Reviewer 1 Report
The paper illustrates an experimental protocol combined with a smart computer mouse for occupational stress assessment. The process is well documented and can be reproduced and verified. The analysis of physiological signals and statistical method are extensively discussed, and appear to be effective in detecting the users' stress levels while working on the computer. Therefore, I would recommend this manuscript to be published, after addressing the following issue:
As PPG signals suffer from motion artifacts and ambient light noise, especially when the subject is moving the mouse. In order to remove these noises, the authors use Kalman filter. However, the BPM data shows large deviations after Kalman filtering (Figure 4). Is the data after filtering reliable? The authors should provide some information on this issue, because it is critical for real applications.
Reviewer 2 Report
The presented manuscript is undoubtedly a relevant scientific work. However, several remarks should be considered.
1. The article’s contribution is unclear in the “Introduction” section. The authors need to formalize the research objective in the “Introduction” section and indicate how this work satisfies or achieves it.
2. The manuscript has no conclusion. Thus, it must be added with the following information: 1) numerical results obtained in work, 2) limitations of the proposed approach, and 3) prospects for future research.
In sum, the submitted manuscript can, in principle, be accepted after minor revisions based on the reviewer’s comments. The authors need to describe a point-by-point response or provide a rebuttal in case some of the reviewer’s comments cannot be revised.
Reviewer 3 Report
Summary:
The authors present a system and an experimental study to detect occupational stress using physiological signals
General concept comments:
While the authors show that BPM and SC population values showed significant statistical differences between the control and stress phases for one or both tasks of the experiment, it is not clear how this translates to individual participants' stress detection. The authors should discuss this in the paper. The authors should also provide a box-plot (or equivalent) of the BPM and SC values across individuals collected in the experiments and discuss the span of the data (the range of values in Figure 8 seems to be very small). The sample size seems to be quite small for an experimental study. The authors should consider increasing the sample size or provide justification for selected sample size or modify the paper into a feasibility study. The authors should also provide a sub-group analysis of the results based on gender, and verify if the conclusions still hold true. The authors should discuss confounding factors such as arrhythmias, sweating, etc that will impact stress detection. Overall, the work is scientifically sound with appropriate study design and validation, and I recommend accepting the article after the comments are satisfactorily addressed.
Reviewer 4 Report
The Authors provide an interesting study to validate the use of a multi-sensor system integrated into a computer mouse in the detection of occupational stress in an office environment. Despite the novelty that this work may present, it presents significant shortcomings.
1) Provide a table or a section on related works as well as a discussion in which it is compared with them to highlight the contribution of this work to the state of the art. Thus the work carried out can be framed and valued. For example:
Belk, M., Portugal, D., Germanakos, P., Quintas, J., Christodoulou, E., & Samaras, G. (2016). A Computer Mouse for Stress Identification of Older Adults at Work. UMAP.
In addition, a comparison should be made with the results obtained with wearable devices:
Can, Y.S.; Chalabianloo, N.; Ekiz, D.; Ersoy, C. Continuous Stress Detection Using Wearable Sensors in Real Life: Algorithmic Programming Contest Case Study. Sensors 2019, 19, 1849. https://doi.org/10.3390/s19081849.
2 2) Provide an electrical diagram of the custom mouse developed.
3 3) Provide a schematic or photo of the simulated office work environment.
4 4) Separate between discussion and conclusion (main highlights and contributions) of the work done.
Round 2
Reviewer 4 Report
Most of the comments made on the first version of the manuscript have been considered and the article has improved significantly. Please, include a more complete description in the captions of figures 1 and 4 before being considered for publication.
Author Response
Dear Reviewer,
Thank you very much for your comment. The captions of the figures were modified to address your comment.